# Likely and High-End Impacts of Regional Sea-Level Rise on the Shoreline Change of European Sandy Coasts Under a High Greenhouse Gas Emissions Scenario

**Rémi Thiéblemont [1],\*, Gonéri Le Cozannet [1] , Alexandra Toimil [2] , Benoit Meyssignac [3] and Iñigo J. Losada [2]**

[1]   Bureau de Recherches Géologiques et Minières "BRGM", French Geological Survey, 3 Avenue, Claude Guillemin, CEDEX, 45060 Orléans, France; G.LeCozannet@brgm.fr
[2]   Environmental Hydraulics Institute "IHCantabria", Universidad de Cantabria, Parque Científico y Tecnológico de Cantabria, Calle Isabel Torres 15, 39011 Santander, Cantabria, Spain; alexandra.toimil@unican.es (A.T.); inigo.losada@unican.es (I.J.L.)
[3]   Laboratoire d'Etudes en Géophysique et Océanographie Spatiales "LEGOS", Université de Toulouse, CNES, CNRS, UPS, IRD, 14 Avenue Edouard Belin, 31400 Toulouse, France; Benoit.Meyssignac@legos.obs-mip.fr
\*   Correspondence: r.thieblemont@externe.brgm.fr

**Abstract:** Sea-level rise (SLR) is a major concern for coastal hazards such as flooding and erosion in the decades to come. Lately, the value of high-end sea-level scenarios (HESs) to inform stakeholders with low-uncertainty tolerance has been increasingly recognized. Here, we provide high-end projections of SLR-induced sandy shoreline retreats for Europe by the end of the 21st century based on the conservative Bruun rule. Our HESs rely on the upper bound of the RCP8.5 scenario "likely-range" and on high-end estimates of the different components of sea-level projections provided in recent literature. For both HESs, SLR is projected to be higher than 1 m by 2100 for most European coasts. For the strongest HES, the maximum coastal sea-level change of 1.9 m is projected in the North Sea and Mediterranean areas. This translates into a median pan-European coastline retreat of 140 m for the moderate HES and into more than 200 m for the strongest HES. The magnitude and regional distribution of SLR-induced shoreline change projections, however, utterly depend on the local nearshore slope characteristics and the regional distribution of sea-level changes. For some countries, especially in Northern Europe, the impacts of high-end sea-level scenarios are disproportionally high compared to those of likely scenarios.

**Keywords:** sea-level rise; high-end; shoreline retreat; projections; Europe

## 1. Introduction

The rising of the global mean sea-level (GMSL) is observed since the early 20th century and is projected to continue and further accelerate over the 21st century [1,2], hence posing a major challenge for coastal regions worldwide [3]. Since the 1970s, the ocean thermal expansion and melting of land glaciers, largely caused by the anthropogenic global warming [4], are the main contributors to the GMSL rise. While the thermal expansion is expected to continue increasing over the 21st century, the total contribution of ice mass loss by ice-sheets is projected to become more substantial, and it is the first driver of the GMSL rise acceleration since 1993 [5–7]. The range of projections of the GMSL by the end of the 21st century is very large, however, namely due to high uncertainties in the understanding of physical processes that drive components of the GMSL and uncertainties in future greenhouse

gas emissions. Since the release of the IPCC AR5 (Intergovernmental Panel on Climate Change Fifth Assessment Report) in 2013, the debate on long-term projections of GMSL has strongly focused on the potentially very large contribution of the Antarctica ice-sheet [2,8–11], which constitutes a deep source of uncertainty.

Sea-level changes at the regional scale can substantially differ from GMSL change. Thermal expansion is modulated regionally by changes in ocean circulation, density, and atmospheric pressure [12,13]. In addition, water mass transfer from land to the ocean—due, e.g., to mountain glaciers and ice-sheets melting or groundwater extraction—induces regional changes by altering the Earth's gravity field, Earth rotation, and solid-Earth deformation [14]. Ongoing changes in the solid Earth are also still caused by the viscous adjustment of the mantle to the important mass redistribution that followed the Last Glacial Maximum (named Glacial Isostatic Adjustment; GIA) [15]. At the European scale, the influence of GIA is particularly prominent in the Scandinavian area. Finally, at a local scale, high-resolution oceanic processes [16] and vertical ground motion [17,18] can affect further the relative sea-level.

Projections of future regional sea-level are crucial to support adaptation planning. So far, coastal adaptation practitioners have often relied on IPCC sea-level projections, which are provided in the form of a likely range (probability larger than 66%), and do not reflect the whole range of uncertainties of sea-level projections [19–23]. However, as coastal climate services are being developed [24,25], it is increasingly recognized that different types of sea-level projections are required depending on the degree of uncertainty tolerance of decision-makers [26]. If the uncertainty tolerance is medium to high, one can use probabilistic projections that are particularly suited to identify the adaptation alternative that has the best-expected outcome [26,27]. In contrast, in the case of high-risk aversion, the uncertainty tolerance is low and, therefore, robust decision-making could be required [26]. Probabilistic projections cannot be used within robust decision-making approaches because ensuring robustness implies testing adaptation options against any plausible scenarios, whereas the tail of sea-level projections is highly uncertain [22,23,28,29]. Instead, one should consider high-end projections or scenarios, which explore plausible—although unlikely—high impact sea-level scenarios beyond the likely range [28]. In particular, high-end sea-level scenarios are particularly useful to plan the full range of coastal adaptation responses [26,27,30,31].

Cascading effects from sea-level rise to coastal impacts are generally quantified with the aim of computing the most likely impacts [32,33]. In fact, previous studies that have addressed future high-end sea-level changes at the regional and local scale [34–36], generally did not assess subsequent coastal impacts on erosion consistently, except in some locations. For instance, Jiménez et al. [37] projected the sea-level rise-induced erosion on the Catalan coast and quantified the impacts on beach functions. Multiple challenges pertain to the evaluation of high-end or likely shoreline changes. First, there is no unique approach to quantify future sea-level rise impacts on shoreline retreats, as shown for example for the coast of Asturias [38], Balearic Islands [39], or Black Sea beaches [40]. Furthermore, in addition to sea-level rise, multiple natural and anthropogenic processes are acting at different space and time scales, and contribute to the observed shoreline changes [41,42]. Finally, the validation of prospective shoreline change modeling frameworks remains challenging because of the observed variability of shoreline changes. For example, it has been estimated that half of the world's sandy beaches are stable, one quarter is accreting, and a remaining quarter is eroding [43], in particular, due to human interventions [44]. Nevertheless, in the climate change context, sandy shorelines will be altered by waves, storm surges, tides, and river flows [38,45–48], whose changes have either not fully quantified effects at European scale or have fewer impacts than projected sea-level rise [49]. As a consequence, sea-level rise is the most common coastal impact of climate change considered in European regulations, engineering designs, and adaptation plans [50,51]. Shoreline retreat projections induced by a high-end sea-level rise at the European scale and for each individual European country has, however, not been quantified yet.

The present study provides a first estimate of the contribution of sea-level rise for shoreline change of European sandy coasts for likely and high-end sea-level rise scenarios by 2100 under business-as-usual greenhouse gas emissions. Our study focuses on beaches, therefore not considering other vulnerable coasts such as non-consolidated cliffs, artificialized coasts and wetlands, and their potential permanent inundation. To do so, we design high-end sea-level scenarios at the European scale that we combined with a database of European coastal settings (EUROSION, 2004) and a recently released nearshore slope dataset [52]. Our sea-level projections account for all relevant regional sea-level component (i.e., sterodynamic, land-water mass transfer, etc.), except local contributions such as local vertical ground motion or high-resolution ocean changes, which needs to be assessed locally [36]. We build two high-end sea-level scenarios, informed by the most recent literature, following two distinct approaches. The first relies on the upper bound of the likely range, while the second follows a "worst-model" approach. Finally, we deliver separate shoreline change projections and beach area loss for each European coastal country.

## 2. Materials and Methods

In this section, we first describe the sea-level change dataset and methods for calculating regional sea-level projections and then the EUROSION database and procedure to derive shoreline change projections of European sandy coasts.

### 2.1. Regional Sea-Level Rise

Regional sea-level changes ($\Delta RSLC$) result from the sum of the global and regional sea-level change components. Considering the inverse-barometer correction [53], $\Delta RSLC$ is expressed as

$$\Delta RSLC = \Delta Z + \Delta R_{barystatic-GRD} + \Delta R_{GIA} \tag{1}$$

where $\Delta Z$ is the sterodynamic induced sea-level change, $\Delta R_{barystatic-GRD}$ is the barystatic-GRD-induced sea-level change (excluding GIA) (GRD: changes in Earth Gravity, Earth Rotation and viscoelastic solid-Earth Deformation) and $\Delta R_{GIA}$ is the GIA-induced sea-level change (GIA: Glacial Isostatic Adjustment). Note that the definition of relative sea-level change (following the terminology of Gregory et al. [53]) further considers local vertical ground motions, which are, however, neglected in our study. The details of the terms of Equation (1) are described in the following subsections.

### 2.1.1. Sterodynamic Sea-Level Change ($\Delta Z$)

The sterodynamic sea-level change corresponds to the sum of the global-mean thermosteric sea-level rise due to the thermal expansion of the ocean in response to global warming, and ocean dynamic sea-level change (which includes the inverse barometer correction). Dynamic sea-level change accounts for long-term ocean circulation changes (e.g., Atlantic meridional overturning circulation (AMOC)) and changes in the density. In this study, the sterodynamic contribution to future sea-level rise is computed from atmosphere and ocean general circulation model (AOGCMs) outcomes of the CMIP5Coupled Model Intercomparison Project (CMIP5), which are provided with a 1°/1° resolution by the Integrated Climate Data Center (ICDC) of the Hamburg University [54]. Recently, the evaluation of a subset of CMIP5 models revealed an overall agreement at the regional scale between the simulated sea-level and tide gauge records over the twentieth century [55].

Our study focuses on coastal zones, where discrepancies in the spatial coverage of AOGCMs that participated in CMIP5 are particularly pronounced. If not accounted for, these coastal coverage discrepancies can thus introduce biases when computing multi-model statistics, which in turn could significantly alter the estimate of shoreline retreat. Hence, the sterodynamic sea-level change for each EUROSION segment (see Section 2.2) is assigned by selecting the nearest ocean grid cell that is covered by all models, except for the semi-enclosed seas, where a specific strategy is used (see Section 3.1). The distances between EUROSION segments and the nearest grid cell do not exceed 300 km except in

the Channel and Gibraltar, where maximum distances of 400 km are found (not shown). This source of uncertainty is acceptable given the broad scale of the CMIP5 simulations available.

### 2.1.2. Barystatic-GRD Induced Sea-Level Change ($\Delta R_{barystatic-GRD}$)

This term accounts for ongoing sea-level changes induced by the mass change $\delta M$ in any of the stores of water on land (i.e., land glaciers, ice sheets, and land water storage). Water mass transfer from the land to the ocean has a global effect due to the addition of water mass to the ocean (barystatic sea-level rise) and a regional effect through instantaneous changes in the geoid (GRD-induced relative sea-level change). Note that the latter effect leads to a redistribution of the regional sea-level globally but does not cause a global-mean sea-level rise. Both contributions are combined into a constant geographical pattern called barystatic-GRD fingerprint $\phi$, [53,56] and are proportional to the land water mass change $\delta M$. The sum of the barystatic-GRD relative sea-level change due to all the stores $i$ of water on the land is expressed by, [53],

$$\Delta R_{barystatic-GRD} = \sum_i \delta M_i \phi_i \qquad (2)$$

Note that water mass change $\delta M_i$ can be instead expressed as a global sea-level equivalent change $\Delta SLE_i = \delta M_i / (\rho_f A)$ where $\rho_f$ and $A$ are the water density and the area of the global ocean, respectively [53]. In practice, $\Delta SLE_i$ gives the global sea-level change contribution of each store of water on the land, and its multiplication by the corresponding fingerprint gives its regional distribution. The land ice contribution includes land glaciers and the ice sheets on Greenland and Antarctica. The ice sheet contributions are separated into a surface mass balance contribution (SMB) and a dynamical process contribution (DYN) because they respond differently to climate change [57]. We also consider projections of change in groundwater and land water storage. Details on all these contributions in the likely and high-end framework are provided in Section 3.

### 2.1.3. GIA-Induced Relative Sea-Level Change ($\Delta R_{GIA}$)

The GIA term accounts for ongoing changes in the solid Earth caused by past changes in land ice, which presently are dominated by deglaciation following the Last Glacial Maximum (LGM) [53]. Decreasing in the mass load on land induces an uplift of areas beneath former ice sheets and subsidence of areas adjacent to former ice sheets. In addition, increasing the mass of the ocean triggers the rising of coastal land and subsidence of the seafloor. Although the GIA influence on the relative sea-level change is small in most low-to-mid latitudes regions, its effect becomes substantial close to former ice sheets of the LGM, such as in Scandinavia. Here, GIA projections and their uncertainties rely upon the combination of two GIA fields that are independently derived from the ICE-5G model [58] and the updated Australian National University (ANU) [59] model. Following the procedure of the IPCC AR5, the GIA mean-field is defined as the mean of the two fields, and "one standard error of the GIA uncertainty is evaluated as the departures of the two different GIA estimates from their mean value" [1].

### 2.2. Shoreline Changes

Shoreline changes projections are estimated by applying the Bruun rule to coastal segments described in the EUROSION database. Since we focus on the impacts of sea-level rise only, shoreline changes caused by processes unrelated to sea-level rise are not included in our projections and should be considered for local assessment.

### 2.2.1. The EUROSION Database

EUROSION is a pan-European coastal database (https://www.eea.europa.eu/data-and-maps/data/geomorphology-geology-erosion-trends-and-coastal-defence-works) built to support the assessment of coastal erosion status and trends. This database is comparable, but independent from state of

the coastal art database used in regional to global coastal impacts models such as the Dynamic and Interactive Vulnerability Assessment (DIVA) framework [60–62]. Specifically, the EUROSION database provides fourteen data layers of factors deemed to impact coastal erosion processes such as geomorphology and geology, tidal regime, wave and wind climate, or sea-level rise throughout Europe. The European coastline, which extends over a length of 130,000 km in EUROSION, is divided into about 50,000 segments of uniform geology and geomorphology.

Previous research exploring the overall consistency of the database concluded that the database displays consistent patterns for coastal geomorphology, geology, shoreline changes, sea-level changes, and wave climates as a whole [63], although some minor discrepancies in the interpretation of the observed shoreline changes can be identified in some places [64]. Here, we rely on the description of geomorphology and geology only, which belongs to the most trustful elements in the coastal database. The geomorphology data layer of EUROSION is an update of the Corine Erosion dataset. Data were collected from a survey addressed to the coastline management department of each European country. In EUROSION, coastline segments are initially defined according to 20 geo-morphological types (e.g., beach with rocky foreshore, beach with sandy foreshore, pocket beach, estuary, hard-rock cliffs, etc.), that we combined into five broader categories (Table 1): sandy beaches, hard-rock shore, soft-rock shore, muddy shore, and artificial shore. As shown in Table 1, about 23% of European coastline (i.e., 30,000 km) geomorphology is not documented; these concerns, namely, a large part of the European Black Sea coastline, Cyprus almost completely (more than 90% of the coastline), and small (but numerous) islands located in the Baltic Sea. In this work, we consider sandy beaches—that cover more than a quarter of the European coastline (Table 1)—and estimate the shoreline retreat based on the Bruun rule.

**Table 1.** Distribution of geomorphology of the EUROSION segments expressed in kilometers and percent of the total European coastline length. See the text for details.

| Geomorphology | Coastline Length (in km) | Coastline Ratio (in %) |
|---|---|---|
| Sandy shore | 34,661 | 26.6 |
| Hard-rock | 35,720 | 27.4 |
| Soft-rock | 11,832 | 9.0 |
| Muddy | 11,436 | 8.8 |
| Artificial | 7252 | 5.6 |
| N-D | 29,405 | 22.6 |
| Total | 130,031 | 100 |

2.2.2. Shoreline Changes Induced by Sea-Level Rise

The Bruun rule [65] predicts the upward and landward displacement of a coastal profile due to sea-level rise in a strict two-dimensional sense as follows

$$\Delta S = \Delta RSLC / tan(\beta) \tag{3}$$

where $\Delta RSLC$ is the regional sea-level change, and $tan(\beta)$ is the slope of the active profile from the depth of closure to the top of the upper shoreface.

Thus, its description of the actual erosion of physical processes is limited and debatable [41,66]. The Bruun rule underlying assumptions include considering that sediment transport only occurs perpendicularly to the shoreline, thus neglecting any tri-dimensional variability, and assuming that the coastal profile is an equilibrium profile that has uniform sediment size. In addition to these assumptions that limit its applicability, the Bruun rule requires two input parameters that have significant uncertainty. On the one hand, the quantity of sea-level rise, whose inherent uncertainty would be unavoidable even if a more sophisticated method were to be used; on the other hand, the nearshore slope, which depends on the active profile width and height above the depth of closure [67]. Due to the lack of field data at many coastal locations, the depth of closure is often calculated with empirical formulations, producing Bruun rule recession estimates that could vary up to 500% [68].

However, there are also reasons in favor of applying the Bruun rule for qualitative first-pass regional-scale assessments. While local shoreline change estimates require using comprehensive sediment budget approaches (e.g., including other processes such as longshore sediment transport), global estimates are more precise because they average across large areas [61,69]. Furthermore, using complex morphodynamical models in broad continental and global scale assessments may be impracticable.

In the absence of global detailed information on nearshore slopes, previous global or continental-scale analysis of erosion using the Bruun rule relied on a constant uniform nearshore slope value assumed to be 1% [61], which corresponds to the average estimate for sandy coasts [70]. Very recently, however, by combining global topo-bathymetric data and global wave reanalysis, Athanasiou et al. [52] released the first global dataset of nearshore slopes spaced at 1 km along the global OpenStreetMap (OSM) coastline (the dataset is available at https://doi.org/10.4121/uuid:a8297dcd-c34e-4e6d-bf66-9fb8913d983d). Using a closest-neighbor procedure, we interpolated the Athanasious et al. nearshore slopes on the EUROSION sandy coastal segments. The results are shown in Figure 1. It reveals an important spread all along the European coastline, with qvnearshore slope ranging from less than 0.1% to more than 50%. Overall, sandy segments in Southern Europe (namely the Mediterranean basin) show far steeper nearshore slopes than those of Northern Europe, especially in the Baltic and North Seas region.

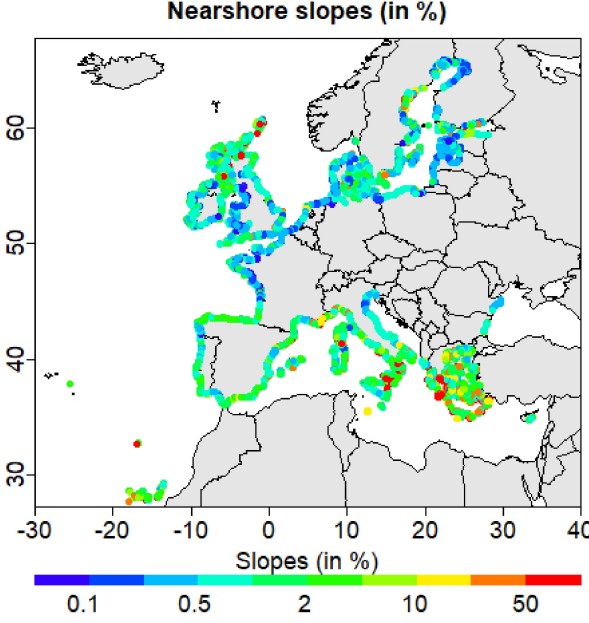

**Figure 1.** Nearshore slopes (in %) from the Athanasiou et al. (2019) dataset interpolated in EUROSION.

In this paper, we present a first-pass assessment of the effects of regional sea-level rise on the shoreline change of the European sandy coasts considering three sea-level rise scenarios (i.e., the likely RCP8.5 and high-end A and B). We apply the Bruun rule to obtain average shoreline change rates in each segment of the EUROSION database containing sandy coasts, to which we associate the closest ocean grid cell of regional sea-level change projections (see Section 2.1). For each EUROSION segment, the nearshore slope is prescribed from the Athanasiou et al. dataset (Figure 1). The 1% constant uniform slope definition [61] is also used for the sake of comparison with previous pan-European assessments [71] and to test the sensitivity of shoreline change estimates to the nearshore slope definition. We provide shoreline change estimates at European and country levels by 2100 with respect 1986–2005. It is important to note that the magnitudes reported only account for one component of shoreline change rather than the total shoreline change that will occur by the end of the century due to all drivers. This component considers the relative sea-level rise in a case of high greenhouse gas emissions only, and it should be interpreted as a broadly indicative estimate.

## 3. Approach for the Assessment of High-End Coastal Impacts

High-ends are defined as plausible—although unlikely—high-impact sea-level scenarios [28]. There is still no unique approach in the sea-level literature on how to quantify high-ends today. In fact, different lines of evidence can be used to define potential future contributions and to combine them [28]. High-end scenarios are often designed in a probabilistic frame based on: (i) Representative Concentration Pathway (RCP) scenarios of the IPCC, (ii) assumptions with regard to physical processes to be considered (e.g., drivers of Antarctic ice-sheet melting) and (iii) particular subsets of models [72]. High-end projections are also informed based on expert elicitations [73,74], whether within a probabilistic approach or not [75].

Here, since we focused on high impact scenarios of high greenhouse gas emissions, our sea-level projections rely on the RCP8.5 emission scenario and its likely range. We deliver two high-end scenarios; the high-end A which is less pessimistic, defined based on the upper limit of the RCP8.5 likely range, while the high-end B follows a "worst-model" approach, that is, not necessarily the upper limit to sea-level rise, which may exceed current modeling outcomes. Details on the high-end A and B scenarios design are given in the following two subsections.

### 3.1. Sterodynamic Component: CMIP5 Model Selection and High-End Definition

Figure 2 shows the sea-level change RCP8.5 projections of CMIP5 models, as provided by the ICDC data center, at the end of the 21st century and calculated for the seven ocean basins surrounding the the European coastline (see Appendix A). The model spread is particularly pronounced in the Atlantic sector, with sterodynamic changes projection ranging from 0.3 to 0.7 m. Despite the overall consistency in the magnitude of model projections at the European scale (e.g., MIROC5 and ACCESS1-0 climate models are found in the upper tail of all distributions), we note that there are significant changes in model ordering between basins. This reflects the regional influence of ocean dynamics and circulation changes on sea level.

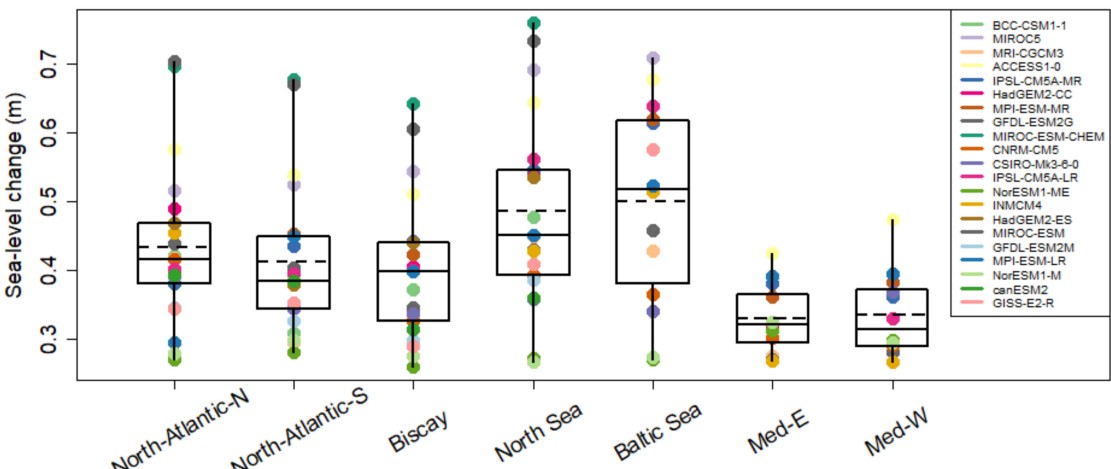

**Figure 2.** Sterodynamic (with applied inverse barometer correction) sea-level projected change in 2099 with respect to the average over the period 1986–2005 for the North-Atlantic-N, North-Atlantic-S, Bay of Biscay, North Sea, Baltic Sea, Mediterranean-E, and Mediterranean-W Sea of atmosphere and ocean general circulation models (AOGCMs) that simulated the Representative Concentration Pathway 8.5 (RCP8.5) scenario as initially provided in the Integrated Climate Data Center (ICDC); i.e., before our model selection. Whisker boxes display the multi-model 1st quartile, median, and 3rd quartile. The horizontal dashed lines indicate the multi-model mean.

Among the 21 models, MIROC-ESM and MIROC-ESM-CHEM project anomalously large sea-level rise in the Atlantic and North Sea areas. If these two models are discarded, the distribution obtained by the 19 remaining CMIP5 models in these areas is no longer significantly different from a Gaussian distribution according to the Shapiro–Wilk normality test. Furthermore, by 2100, the global-mean thermosteric sea-level rise of these two models (0.5 m for the RCP8.5 scenario) exceeds the median global-mean thermosteric sea-level rise of all other models (0.3 m) beyond 5 sigma [76]. Finally, the CMIP5 historical MIROC-ESM and MIROC-ESM-CHEM simulations revealed unrealistic sea-surface height values of -15 m in the Mediterranean area that may suggest important biases in the regional sea-level calculations in these two models [77]. MIROC-ESM and MIROC-ESM-CHEM were, therefore, removed from our model selection.

Figure 2 also shows that semi-enclosed seas are not fully covered by all models—among the 21 models, only 14 and 12 cover the Baltic and Mediterranean basins, respectively. These differences between model spatial coverage result in inconsistencies when computing multi-model ensemble statistics, which in turn could significantly affect the spatial homogeneity of regional sea-level rise projections and hence alter their credibility. Furthermore, the rather coarse resolution of AOGCMs prevents an accurate representation of small-scale processes, for example, the water exchange at Gibraltar, which in turn affects regional sea-level estimates in marginal seas [78,79]. Hence, to remove these potential sources of errors, the Mediterranean sterodynamic sea-level projections are constrained with those of the Atlantic area near Gibraltar, which is the Mediterranean Sea entry point. In a similar way, the Baltic Sea projections are calculated using those of the North Sea. This procedure leads to an increase in the consistency of multi-model statistics compared to those relying on the CMIP5 models within the semi-enclosed basins [54,57,80] and is also more realistic with regard to the processes governing multi-decadal sea-level changes in these marginal seas [81–83]. Finally, our selection choice leads to an increase in the Mediterranean projections compared to previous assessments based on AR5 data.

Hereafter, we follow the IPCC AR5 method to define the "likely-range" and therefore compute it as the standard deviation interval around the multi-model mean multiplied by 1.64. For the two high-end scenarios, sterodynamic contributions for each grid cell are defined using the upper bound of the multi-model likely-range (High-end A) and the multi-model outcome maximum value (High-end B, worst model estimate).

## 3.2. Barystatic-GRD Components

Barystatic-GRD induced contributions to European sea-level change projections are defined based on sea-level barystatic-fingerprints computed from data of the ICDC data center, which we multiply by the global sea-level equivalent for each component [54,57,80,84]. The global sea-level equivalent contribution corresponds to the change in the sea-level induced by the transfer of water mass from land water storage to the ocean. As shown in Table 2, the high-end A scenario—corresponding to a "moderate" scenario—is prescribed based on the barystatic component projections of the upper limit of the RCP8.5 global warming scenario "likely-range" [1], except for Antarctica ice-sheet dynamics contribution which experienced several updates since 2013 as summarized in the recent IPCC Special Report on Ocean and Cryosphere in a Changing Climate (SROCC) [2]. For the high-end B—a more pessimistic scenario—the "worst model approach" was followed.

In the AR5, projections of sea-level equivalent for glaciers were estimated based on outcomes of global glacier models forced by temperature and precipitation projections from RCP scenarios simulations by climate models [85–87]. Revisions of existing projections (e.g., Marzeion et al., 2012 [86]) and new modeling estimates [88] resulted in slightly lower glacier mass losses [79]. Nonetheless, given the limited evidence of substantial changes in glacier mass loss projections since the AR5 [2], glacier contributions for the high-end A scenario are defined as the upper limit of the likely range, which corresponds to a sea-level equivalent of 0.26 m. Following the "worst-model approach" glacier sea-level equivalent (SLE) for the high-end B is set to 0.29 m, which is the maximum estimate obtained

by Marzeion et al. (2012) [86] when forcing their glacier model with CMIP5 simulation outputs of the Hadley Global Environment Model 2—Earth System (HADGEM-ES).

AR5 projections of land water account for groundwater depletion [89], which contributes to sea-level rise, and a negative contribution through increasing land water storage due to dams over the 21st century [90]. Although Wada et al. (2016) [91] showed recently that previous studies might have inflated groundwater depletion contribution to sea-level rise since full groundwater drainage to the ocean was assumed without accounting for pumped water remaining on land (~20%), the AR5 projections have not yet been re-assessed substantially. Therefore, in the absence of worst-case model estimates in the literature (to our knowledge), both high-end scenarios thus rely on the upper limit of the likely range provided in the AR5 and SROCC (0.11 m).

**Table 2.** Global mean sea-level changes by 2100 relative to 1986–2005 of each barystatic-GRD contribution for (left) the AR5/SROCC median and likely range (in brackets), (middle) the high-end scenario A and (right) the high-end scenario B. All values are rounded at two significant digits beyond the decimal point. See the text for details.

| Component | RCP8.5 IPCC AR5/SROCC | High-End A Moderate | High-End B Worst Model |
|---|---|---|---|
| Glaciers | 0.18 [0.10 to 0.26] m | 0.26 m | 0.29 m [2] |
| Greenland [1] | 0.15 [0.09 to 0.28] m | 0.28 m | 0.34 m [3] |
| Antarctic SMB | −0.05 [−0.09 to −0.02] m | −0.02 m | 0 m [4] |
| Antarctic DYN | 0.16 [0.02 to 0.37] m | 0.37 m | 0.8 m |
| Groundwater | 0.05 [−0.01 to 0.11] m | 0.11 m | 0.11 m |

[1] Greenland estimates combine surface mass balance (SMB) and dynamic effects (DYN). [2] Marzeion et al. [86]. [3] 0.34 m corresponds to two times the worst model case in Fürst et al. [92]. [4] This scenario assumes no aggravation of accumulation over the Antarctic ice-sheet.

Greenland ice-sheet contribution to sea-level change is driven by changes in surface mass balance (SMB) and dynamic effects (DYN). IPCC AR5 [1] (and unchanged in SROCC [2]) estimated that, by 2100, the upper limit of the likely range of Greenland's SLE contribution would be 0.28 m under the RCP8.5 scenario, dominated by SMB by two thirds. This value defines our high-end A scenario. Recently, Fürst et al. (2015) [92] forced an ice-sheet model with 10 AOGCMs from the CMIP5 dataset and found a slightly smaller median contribution of Greenland to global sea-level rise by 2100 than in the AR5 (0.10 m versus 0.15 m). Their highest Greenland SLE projection (0.17 m by 2100) was found when the ice-sheet model was forced by canESM2 RCP8.5 simulation results. However, using also a subset of CMIP5 models (including canESM2) and comparing with reanalysis, Delhasse et al. (2018) [93] have shown that some changes observed over the last two decades in atmospheric circulation (i.e., increase in blocking high frequencies in summer) could not be reproduced by the CMIP5 models. They further concluded that, if the current summer atmospheric circulation pattern over Greenland happens to persist, projected Greenland SMB contribution to sea-level rise could be amplified by a factor of two for a similar temperature increase. Therefore, we build our high-end B scenario by considering that the largest model projection found in Fürst et al. (2015) [92] could be amplified by a factor two following the arguments of Delhasse et al. (2018) [93]; this would lead to a Greenland SLE of 0.34 m by 2100. Note that the most recent expert elicitations on ice-sheet contribution suggested an upper bound of the likely range at 60 cm [73], which gives us confidence that our high-end B Greenland component projection remains credible.

Since the release of the AR5, the Antarctica ice-sheet contribution has been highly debated [2]. It could be one of the most important contributions to future sea-level rise, in particular under a high global warming scenario. The uncertainties on this contribution are, however, very large and strongly depend on the understanding of Antarctica ice-sheet dynamic processes and their evolution under a warming climate. Lately, two mechanical processes that may trigger important dynamic mass loss of the ice sheet have been intensively discussed. First, the marine ice sheet instability

(MISI), which is probably observed already in West-Antarctica [94,95], and second, the marine ice cliffs instabilities (MICI), more hypothetical than MISI (and never observed over the instrumental period), which involves a rapid retreat of ice shelves through hydrofracturing and subsequent collapse of ice cliffs formed at the ice sheet margins [8]. Although the effectiveness and magnitude of these two mechanisms remain to be determined, considering their potential contribution to SLR strongly inflates the likely range provided in the AR5. Therefore, the median contribution of the Antarctica ice-sheet dynamics has been recently re-assessed to 0.16 m in 2100 under the RCP8.5 scenario in the SROCC [2] (against 0.08 m in the AR5). Note that among the studies (five in total) considered in the SROCC to retrieve this new estimate, some relied on simulations that included MISI (but not MICI). The large spread in the projections of these five studies further led to extend the upper bound of the RCP8.5 likely range substantially from 0.19 m (AR5) to 0.37 m (SROCC) by 2100. The latter value is hence used for the high-end A scenario. For the high-end B, we consider a mean projection assuming MICI (and not a worst-case model outcome) because the confidence in MICI projection is still debated, and it is unsure that it will be initiated over the 21st century. In their former paper, DeConto and Pollard (2016) [8] estimated that MICI could contribute to global sea-level rise to more than 1 m by 2100. Very recently, Edwards et al. (2019) [9] revisited the latter results by considering the full range of uncertainties of the ice-sheet model parameters used by DeConto and Pollard (2016) [8]. This more robust statistical treatment by Edwards et al. (2019) [9] led to revised downward the DeConto and Pollard (2016) [8] projection to 0.8 m. The latter value 0.8 m is hence used for the high-end B.

In contrast with Greenland, the Antarctic ice-sheet SMB is projected to contribute to a drop of global sea-level, namely due to an increase of snow accumulation under future atmospheric warming [96]. Given the absence of new estimates since the AR5 (particularly because no new CMIP simulations are available), the high-end A scenario was designed based on the upper bound of the AR5 likely range. The high-end B assumes no increase of precipitations over the Antarctic.

## 4. Results

### 4.1. Shoreline Changes at the European Scale

Projections of the regional sea-level change off European coasts by the end of the 21st century for the RCP8.5 median, high-end A, and high-end B, are shown in Figure 3 (data are available numerically as Supplementary Materials). Under the RCP8.5 scenario, the median projection is found to be larger than 0.7 m for most of the European coasts except in the Baltic area where a sea-level drop as low as—0.2 m is projected north of ~60°N. In sectors such as the Mediterranean and North Seas, sea-level rise projections exceed 0.8 m. These projections emphasize a substantial contrast of sea-level change projections at the European scale, which is dominated by a gradual northward decrease in the magnitude of sea-level rise. This negative gradient is induced by the sea-level drop contribution within ~2000 km of the Greenland ice sheet and the post-glacial landmass rebound in the Baltic sector, which leads to a relative sea-level drop too.

Under high-end A and B scenarios, sea-level change for most of the European area is projected to be higher than 1.2 and 1.7 m, respectively. Note that our projections lie within the 66% uncertainty bound of global sea-level rise projections under a high emission scenario (5 °C by 2100) as derived from the most recent structured expert judgment [73]. This gives us confidence that our high-end scenarios remain credible, given the knowledge available today. The maximum sea-level change is found in the central North Atlantic Ocean (see western bounds of maps in Figure 3, south of Iceland), where the sea-level rise reaches up to 2.3 m under the high-end B scenario. This Atlantic "hot-spot" has been suggested to be triggered by sea-surface dynamics changes induced by changes in the Gulf stream [80,97,98]. Nearshore, sea-level change projections for all scenarios show a particularly high sea-level rise in the North Sea area, which appears to be robust among CMIP5 climate models (Figure 3).

These pan-European sea-level change projections are characterized by an important regional disparity. This has practical implications for adaptation as more European countries and agencies

are including sea-level scenarios in their regulations or design guidance as part of their adaptation policies [24,27]. For instance, a sea-level rise scenario of 0.6 m by 2100 is assumed for coastal risk prevention plans to limit urbanization in low-lying areas in France [24]. Our results show that under sustained greenhouse gas emissions (i.e., RCP8.5 scenario), this scenario could be possibly doubled (High-end A) or tripled (High-end B) along the France coastline.

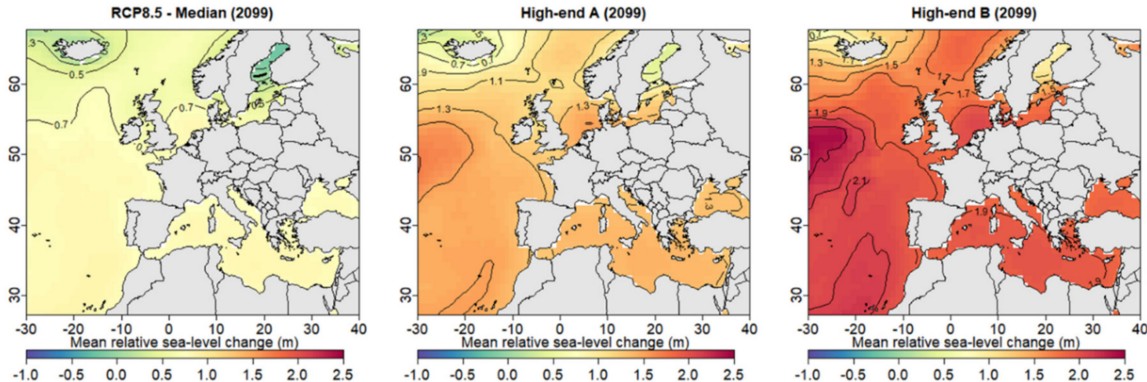

**Figure 3.** Projections of the relative sea-level change (in m) off the European coast by 2100 relative to 1986–2005 for (**left**) the median of the RCP8.5 scenario, (**middle**) the high-end A, and (**right**) the high-end B.

Figure 4 displays the contribution of sea-level changes to coastline changes by 2100, as modeled by the Bruun rule. We first examine the results obtained with a variable nearshore slope (top panel). Regardless of the scenario, the largest projections of SLR-induced shoreline retreat (i.e., greater than 100 m for the RCP8.5 scenario) are mainly found along the North-Atlantic, the North Sea, and Baltic Sea coasts, where the nearshore slope is particularly soft (see also Figure 1). In these regions, the SLR-induced shoreline retreat increases dramatically under high-end scenarios compared to likely projections. Conversely, in the Mediterranean, the overall steeper nearshore slopes lead to reduced shoreline retreat projections (less than 50 m for the RCP8.5 scenario) compared to Northern Europe. Under high-end scenarios, the SLR-induced Mediterranean shoreline retreat is not projected to be amplified except in sector with rather soft nearshore slopes (e.g., East and North-East coast of Italy). Under the RCP8.5 scenario, the Northern Baltic Sea is the less impacted sector by shoreline retreats and even the only region where shoreline progress is projected, due namely to the GIA influence. This, however, does not hold for high-end scenarios: the shoreline of the Baltic states and the Northern Finland appears to be strongly impacted, namely due to the low inclination of the nearshore slope (see Figure 1).

The middle row panel in Figure 4 shows the shoreline change projections by the end of the 21st century when a constant uniform slope of 1% is used instead. These shoreline change projections differ utterly from those obtained with a variable nearshore slope. It appears that the magnitude and spatial distribution of the shoreline change throughout the European coastline evolves linearly with the sea-level projections (i.e., Figure 3), projecting independently of the scenario - maximum shoreline retreats in the Mediterranean, North Sea and Southern Atlantic (i.e., Canary Islands) areas, and minimum retreats (even accretion in the RCP8.5 case) in the Northern Baltic area, as expected from the GIA influence. We, however, notice that the regional contrasts vary between scenarios. The adjusted linear color scale used on Figure 4 (middle row panel) allows understanding this phenomenon: the relative difference of SLR-induced shoreline retreat between the Canary Islands and Eastern Mediterranean increases with more pessimistic scenarios, which in turn implies a reduction of the relative poleward gradient between the Mediterranean Sea and Baltic Sea. In this case, this is primarily because the Canary Islands are very sensitive to the contribution of the Antarctic ice-sheet dynamic component. Our results hence stress that in addition to an important regional contrast at the European scale, the magnitude and distribution of these contrasts vary with future scenarios.

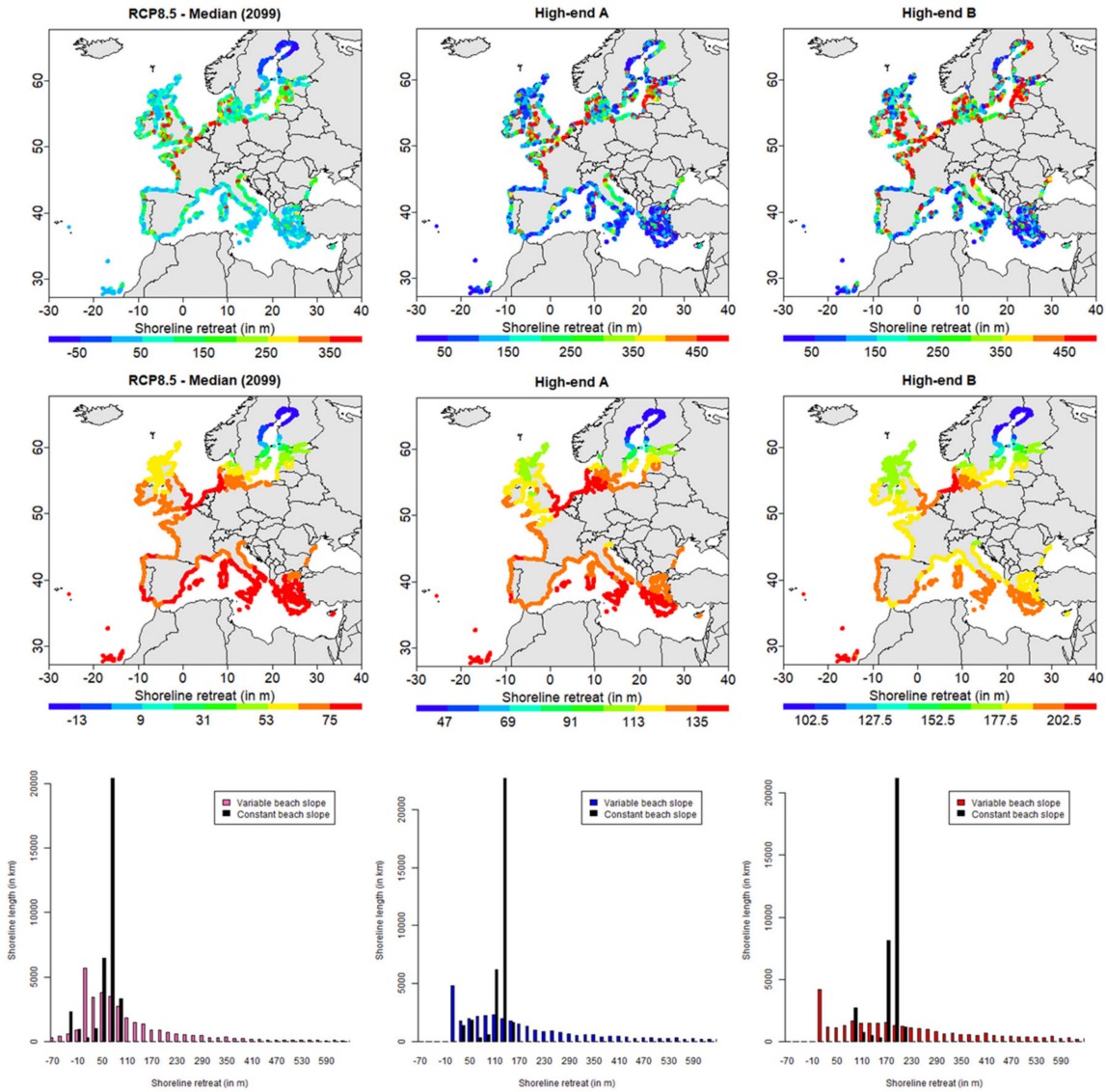

**Figure 4.** Projections of SLR-induced European coastline changes (in m) by 2100 calculated for (**left**) the median of the RCP8.5 scenario, (**middle**) the high-end A, and (**right**) the high-end B. Coastline retreat is displayed for each sandy beach segment of the EUROSION database considering (**top**) variable nearshore slopes provided by Athanasiou et al. (2019) and (**middle**) a 1% constant nearshore slope. (**bottom**) Cumulated length of coastline segments (in km) distributed against their coastline retreat projections (in m) for a variable (colored bars) and constant (black bars) nearshore slopes.

A more quantitative comparison of the two approaches used to estimate SLR-induced shoreline projections is provided by the histograms showing the cumulated length of shoreline as a function of the projected changes per segment (Figure 4, bottom row panel). In the case of the constant uniform nearshore slope approach (black histograms), the shoreline change at the European scale is bimodally distributed, the secondary peak of the distribution owing to the influence of the GIA. Under scenarios with increasing impacts, the distribution is translated toward higher shoreline retreat values. The median values of 70 m, 128 m, and 186 m are found for the RCP8.5, High-End A, and High-End B, respectively. In the case of the variable nearshore slope approach (colored histogram), although the median values are of same order of magnitude (74, 143, and 209 m for the RCP8.5, High-End A, and High-End B, respectively) than in the case of a constant uniform nearshore slope, the distribution shows a much larger spread and is right-skewed. This comparison of the two approaches reveals overall that the regional SLR-induced shoreline retreat projections are very sensitive to both the

nearshore slope and the regional sea-level projections under different scenarios. Next, we examine SLR-induced shoreline change projections at the country scale and further explore the sensitivity to the nearshore slope definition.

### 4.2. Shoreline Changes at Country Scale

The distribution of median SLR-induced coastline retreat per European countries for all scenarios considered in this study is shown on Figure 5. The disparity between European countries in the magnitude of the SLR-induced shoreline change projection is very substantial and further increases under high-end scenarios. For instance, the projected shoreline retreat gap between Greece and Belgium increases from ~300 m to ~900 m between the RCP8.5 median scenario and the high-end B. Overall, countries surrounding the Mediterranean basin (e.g., Greece, Cyprus, Malta, Italy, . . . ) show minimum SLR-induced shoreline retreat projections, which, in addition, do not inflate under high-end scenarios (all projections are found below 200 m). This is primarily due to the steepness of the nearshore slopes in this region compared to the rest of Europe. In contrast, the general weak inclination of nearshore slopes of Northern European countries (e.g., Belgium, Baltic States, Netherlands, Germany,) leads to particularly large SLR-induced shoreline retreat projections, which further strongly amplify under high-end scenarios. It is however important to recall that such potentially huge shoreline retreats are only possible in the absence of any adaptation measure and if the backshore is erodible and low-lying.

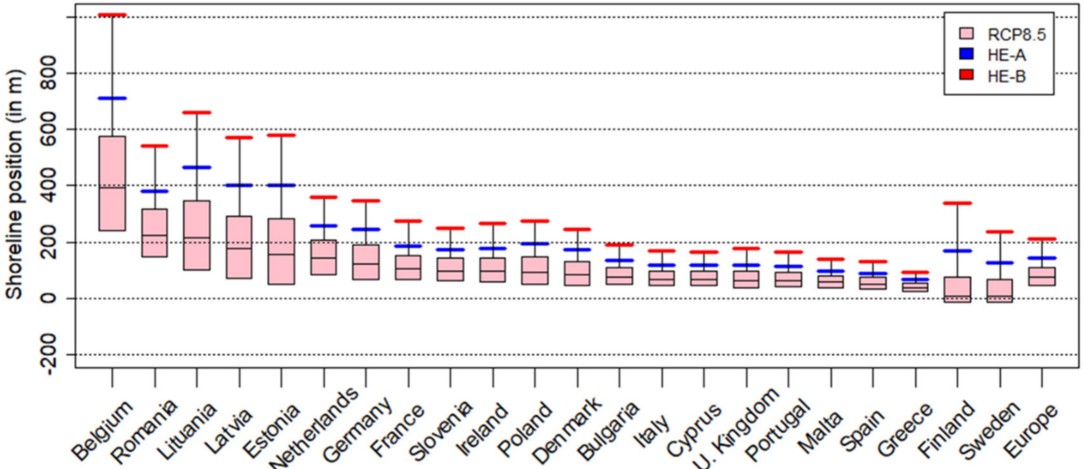

**Figure 5.** Projections of median shoreline retreat by 2099 of sandy coasts, per European country, calculated for (pink) the RCP8.5 likely range, (blue) high-end A, and (red) high-end B. Variable nearshore slope is considered here.

In Figure 5, countries on the x-axis have been ordered from the most to the least impacted according to the RCP8.5 median scenario. This ordering, however, does not hold for both high-end scenarios or even for bounds of the RCP8.5 likely-range. For instance, while Finland and Sweden appear to be the less impacted countries under the RCP8.5 median scenario, they stand among the ten most impacted countries under the high-end B scenario. This effect is namely, due to the dependence of the shoreline change to the inverse of the nearshore slope according to the Bruun rule, which leads to an amplification of SLR-induced shoreline changes for coasts with flat nearshore slopes. Hence, for countries that have coasts with low nearshore slope inclination, the severity of impacts can differ strongly depending on the magnitude of future regional sea-level change.

To better evaluate the impact of the high-end sea-level rise scenarios on the shoreline, we finally estimate the total beach area removal per European country, using the coastal geomorphology information available in the EUROSION database (see Section 2). Results are shown in Figure 6 (top) for both approaches; i.e., variable (solid line) and constant uniform (dashed line) nearshore slope. Here,

the x-axis is organized by increasing order as a function of country coastal length. The total European shoreline loss has an area of 2060 km$^2$, 4140 km$^2$, and 6100 km$^2$ for the RCP8.5-median, high-end A and high-end B scenarios, respectively if the 1% uniform nearshore slope is used, but 4040 km$^2$, 8950 km$^2$ and 13,470 km$^2$, respectively, if the variable nearshore slope is used instead. Besides, a substantial global underestimation of beach removal projections when considering a 1% uniform nearshore slope, the magnitude and distribution of the impact per country appeared to depend on the nearshore slopes definition largely. Indeed, if a uniform slope is used, European countries with longer sandy shorelines, such as Greece, the United Kingdom, Italy, or France, are evidently more affected. Using the variable slope, Greece appears no longer strongly affected (as expected from the steepness of the slopes), while Germany and Ireland then stand among the most impacted countries. As noticed above, the consideration of variable slopes can further exacerbate the dependence of severity of the impact on the various scenario for some countries (e.g., Sweden, Germany, France, United Kingdom, etc.).

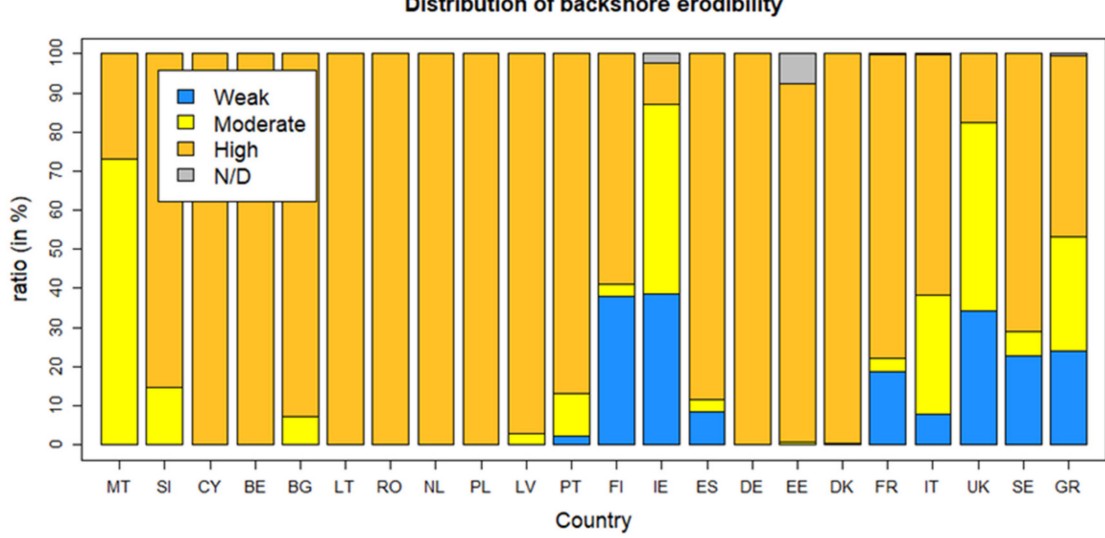

**Figure 6.** (**top**) Projections of the beach area removal per European country in 2099 for the RCP8.5 likely range (pink), the high-end A (blue), and high-end B (red) considering variable (solid) or 1% constant (dotted) nearshore slopes. (**bottom**) Distribution (in %) of the degree of erodibility of the sandy beach backshore for each European country according to the geological layer of the EUROSION database.

### 5. Discussion and Conclusions

In this study, we provided first pass high-end estimates of the sea-level rise contribution to European sandy coastline retreat by the end of the 21st century. High-end scenarios are high-impact and unlikely—but possible—sea-level scenarios that are particularly suited in robust decision-making contexts [26]. Three sea-level scenarios were considered, all built upon greenhouse gas emissions that follow an RCP8.5 trajectory. The first scenario is simply defined by summing the different sea-level contributions and considering the corresponding likely range, as originally published in the AR5 and updated recently in the SROCC. The second scenario (High-end A) relies on the upper bound of the AR5/SROCC likely range. Finally, the third scenario (High-end B) follows a worst-model approach; i.e., we selected the outcome of the model showing the highest sea-level projection for every component based on the most recent literature. The shoreline changes induced by the sea-level rise were estimated from the Bruun rule applied to the EUROSION database sandy beach segments, and for which, we considered either the new nearshore slope dataset provided by Athanasiou et al. (2019) [52] or a uniform slope of 1%, a common approach used for continental-to-global scale shoreline retreat assessments [61,71].

The RCP8.5, high-end A and high-end B scenarios induce, by the end of the 21st century, a relative sea-level rise off European coasts larger than 0.7 m, 1.2 m and 1.7 m, respectively. Nevertheless, these projections feature substantial regional variations, which in turn strongly influence the spatial distribution of SLR-induced shoreline changes in Europe. Our main findings are:

- The magnitude and regional distribution of SLR-induced shoreline change projections by 2100 utterly depend on the nearshore slope, the regional distribution of sea-level changes (i.e., hence the various regional contributions) and the trajectory of the future scenario.
- Ignoring the variability of nearshore slopes and assuming a 1% constant uniform nearshore slope instead may lead to a substantial underestimation of SLR-induced shoreline retreat and beach area removal (reduction by 50%) in Europe. In the absence of any coastal adaptation measure, and assuming an infinite erosion potential of each EUROSION segment, we found that Europe is projected to accumulate a land loss area of 4040 km$^2$, 8950 km$^2$, and 13,470 km$^2$ for the RCP8.5 median, High-end A and High-end B scenarios, respectively.
- The sequencing of countries with respect to their exposure to future shoreline retreat varies very importantly with scenarios. In particular, in Northern Europe, the impacts of high-end sea-level scenarios are disproportionately high compared to those of likely scenarios and to those of the Mediterranean area. This is because the softer the nearshore slope, the more sensitive to sea-level changes in the shoreline. Subsequently, large uncertainties in future regional sea-level changes affect primarily coastal regions with gentle nearshore slopes. This results in substantial changes in the ranking of coastal impacts and adaptation needs in Europe, which may be relevant to consider in adaptation finance mechanisms.

In a previous assessment [71], the European Union land loss rate without adaptation measure was estimated to be of 3.4 km$^2$/year, 6.7 km$^2$/year, 9.9 km$^2$/year and 16.4 km$^2$ /year in 2010, 2030, 2050, and 2100, respectively, considering a climate change scenario leading to a global-mean sea-level rise of 45 cm by 2100 (scenario A2). This corresponds to a European cumulated land loss area of ~1000 km$^2$ over the 21st century. To a zeroth-order approximation, assuming proportionality between land loss and sea-level, a global-mean sea-level rise of 84 cm by 2100 (RCP8.5) would lead to a cumulated land loss of 1870 km$^2$, which, at first glance, appears to be consistent with our results based on the 1% uniform nearshore slope (we found 2060 km$^2$ loss). However, the DIVA model [60,62], upon which these estimates rely, strongly differs from our modeling framework. First, the percentage of erodible sandy shoreline at the global scale in DIVA is about 11% [33,43], while in EUROSION and other independent global databases [43], this percentage is generally larger than 20%. Second, the DIVA model not only accounts for the direct shoreline change predicted by the Bruun rule but also considers the indirect erosion in the tidal basin using a simplified version of the ASMITA model [61]. More

importantly, they found that indirect erosion accounts for more than 50% of the total land loss [61]. In our study, indirect erosion in the tidal basin is not accounted for because the open tidal basins are either not included in the EUROSION database or because they are dominated by muddy coasts, which are not identified as sandy coasts (Table 1). Overall, this indicates that by assuming a 1% uniform nearshore slope, both our and DIVA assessments potentially underestimate the land loss area (in the absence of adaptation measures) and, therefore, likely minimize the subsequent socio-economic impact. Furthermore, as shown in our study, ignoring the local nearshore slope characteristics—at least for Europe—could lead to a large underestimation of the land loss area. This stresses the need to promote further coastal impact models intercomparison exercises (such as COASTMIP) in order to quantify coastal impacts in face of sea-level rise better.

Besides, the above-discussed sources of uncertainty, our shoreline change modeling approach has limitations that may significantly affect our shoreline change estimates. First, the limitations inherent to the Bruun rule are mainly its non-applicability in coasts where processes other than sea-level rise-induced cross-shore sediment transport prevail, and the lack of its comprehensive validation, since sea-level rise-induced shoreline change can be significantly masked by shoreline changes driven by other processes (e.g., sand losses during storms, aeolian transport, alongshore gradients in longshore transport) [67]. Recently, alternative approaches to the Bruun rule have been proposed. They assume that the effects of sea-level rise differ depending on waves and storm climates in each areas and compute sediment losses at dunes toes as sea-level rises [38,68,99–101]. So far, these approaches have delivered lower rates of shoreline retreat than the Bruun rule [102]. Neither the Bruun rule nor the alternative modeling approaches have been convincingly validated yet. However, the fact that the Bruun rule predicts shoreline retreats larger than those of other models suggests it can be used for a high-end estimation of shoreline changes induced by sea-level rise.

Additional limitations of our analysis include: (i) the fact that local vertical ground motion (beside GIA) are not accounted for, (ii) the non-consideration of nearshore slope uncertainties and potential beach slope changes in the future (e.g., through notably to changes in the wave climate [103,104]), (iii) the consideration of beaches only, while other systems (e.g., wetlands) are also vulnerable to sea-level rise, (iv) the fact that we do not assess shoreline changes induced by the permanent inundation of low-lying areas (which however will in practice very much depend on adaptation practices), or (v) the assumption that erosion can continue indefinitely rather than be limited by geological constraints. Regarding the latter point, nonetheless, a qualitative estimate of the geological constraint on the erodibility is provided in the coastline geology layer of the EUROSION database. This layer lists more than 30 types of lithology that we classified with respect to three degrees of erodibility: weak (e.g., granite, basalt), moderate (e.g., limestone rock, sandstone) and high (e.g., sand, loess). The shoreline distribution per country (expressed in %) of the degree of erodibility is shown in Figure 6 (bottom). This qualitative analysis can be viewed as a confidence index of the land loss projections. For instance, Ireland, Greece, and the United Kingdom present a substantial part of the shoreline (more than 50%) that has a weak to moderate erodibility degree. Hence, it is very likely that our high-end land loss projections are overestimated for these countries. In contrast, Germany or Denmark sandy coasts are found to be more prone to erosion as revealed by their large erodibility potential; in this case, our projections are less affected. Finally, as our estimate of shoreline changes only consider erosion of sandy beaches induced by sea-level rise, the actual shoreline retreat induced by sea-level rise could be larger in many low-lying coastal environments. For example, Mediterranean lagoon-type coasts bounded by sandy spits will not only be affected by the erosion of the sandy coast, but also by the permanent inundation of parts of the low-lying coastal plain lying behind the sand spit.

The high-end approach explored in our study is particularly adapted for decision-making applications and adaptation planning, especially for stakeholders with low tolerance to uncertainty [26,28]. Shoreline retreat projections are, however, deeply uncertain, owing to the high diversity of uncertainty sources and their large magnitude. Hence, to reduce uncertainties and improve the reliability of shoreline change projections, it is crucial to improve coastal impact models and regional sea-level

projections. In addition, to account for the full range of uncertainties and quantify their relative weight on future projections, the extra-probabilistic framework appears well suited. This will be the purpose of near-future work.

**Supplementary Materials:** The following are available online at http://www.mdpi.com/2073-4441/11/12/2607/s1. Regional sea-level change projections designed in this study are provided at to be defined.

**Author Contributions:** Project design—G.L.C., I.J.L.; methods—G.L.C., R.T., B.M., A.T.; software for sea-level projections—R.T., G.L.C.; analysis of the results—R.T., G.L.C., writing: All authors.

**Funding:** This research was funded by the BRGM, IH-Cantabria, and the ERA4CS (grant number: 690462). Alexandra Toimil and Iñigo J. Losada were also funded by the Spanish Government through the grant RISKCOADAPT (BIA2017-89401-R). Alexandra Toimil was further funded by Universidad de Cantabria through the 2018 Postdoctoral Fellowship Program.

**Acknowledgments:** This study is part of the COasTAUD framework. We thank Melisa Menendez for useful discussions on the preliminary stages of this study and Julie Billy, Aurélie Maspataud, and Franck Desmazes for their support and recommendations in the analysis of nearshore slopes. We thank Mark Carson for making available the ICDC sea-level data and the modeling groups that participated in the CMIP5 for producing their model output. We also thank the two reviewers for their highly relevant comments and suggestions that contributed to improving this study significantly.

**Conflicts of Interest:** The authors declare no conflict of interest.

## Appendix A. Definition of European Main Basins

Figure A1 displays the seven basins surrounding European coasts that are defined in our study to explore the sterodynamic projections of each of the 21 climate models provided in the ICDC dataset. The following coordinates have been used: Mediterranean-East [12°E–36°E, 29°N–39°N], Mediterranean-West [0°E–12°E, 36°N–45°N], Baltic Sea [14°E–29°E, 53°N–66°N], North Sea [3°W–10°E, 52°N–60°N], Bay of Biscay [9°W–0°E, 44°N–48°N], North-Atlantic North [20°W–5°W, 48°N–60°N], North-Atlantic North [20°W–9°W, 30°N–43°N].

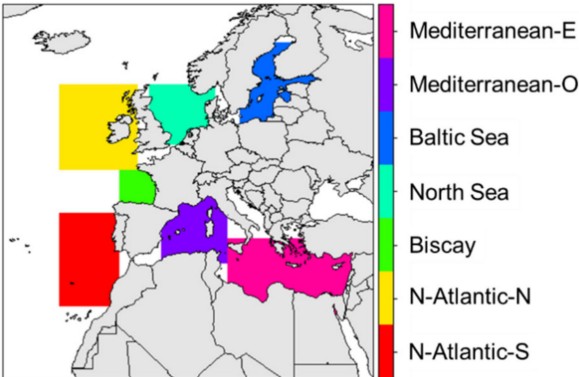

**Figure A1.** Sea area selected to define the Mediterranean-East, Mediterranean-West, Baltic, North Sea, Bay of Biscay, North-Atlantic North and North-Atlantic South basins used to produce Figure 1.

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
