# Peer review of "Likely and High-End Impacts of Regional Sea-Level Rise on the Shoreline Change of European Sandy Coasts Under a High Greenhouse Gas Emissions Scenario"

_water, doi:10.3390/w11122607_

Round 1

Reviewer 1 Report

The manuscript discusses the development of regional sea-level projections for two high-end scenarios for Europe and, based on those scenarios, presents shoreline change projections for European sandy coasts. The manuscript is well written and the methods and results are nicely presented. Despite using simple methods, the authors have provided good justifications for their use and acknowledged and discussed the limitations. However I do have some concerns that would need to be addressed; and some suggestions which may help the authors improve the manuscript. Importantly, the authors have missed two very relevant references, need to extend the discussion on the implications of their results, and need to ensure access to the base data they are using. My points are listed below, in order of appearance (not importance) in the manuscript. I hope the authors find my comments and suggestions useful.

Although the development of the regional sea-level change projections occupies a large part of the manuscript and is one of the outputs of this paper, it is not mentioned in the abstract. I think that this point should also be included in the abstract.

Line 52: why country scale in particular and not local or regional scale?  

Line 54: "do" instead of "does".

Line 69: I am not sure what the authors mean by "...did not assess subsequent coastal impacts consistently...". Are they only referring to erosion? 

Line 82: I think "only" should be removed 

Line 121: "the" spatial coverage   "multi-model" not models 

Line 166: superimposing is probably not good enough as it would not account for non-linear effects. Maybe use a simpler term (e.g. considered)  

-I have not been able to access the Eurosion database. As the authors provide a link, this should provide direct access to the dbase. This is particularly important as there are issues with the database (which the authors do acknowledge) and therefore it should be clear where the input data come from. 

Because of the those issues of accuracy and consistency with Eurosion, it might be useful to compare the dbase with other information to get an idea about its validity. To my knowledge, there are datasets available - e.g. DIVA, which the authors cite, local datasets (if available) or data from crowd sourcing from www.coastwards.org . I am not suggesting a full scale analysis here but just some comparisons to get an impression about the quality  

Line 213: "global estimates are expected to be more accurate as they average across large areas". This is a statement which is repeated often by many and in very different contexts -  I find this rather misleading. Please explain or remove

I find it a bit confusing that section 3 is separated from 2 as it seems to belong to methods, unless I have misunderstood something

Very important: A paper providing coastal slope estimates globally has very recently been published: "Global distribution of nearshore slopes with implications for coastal retreat" Panagiotis Athanasiou et al. 2019 Earth System Science Data. I would assume that it would not be easy (or desirable) for the authors to repeat the analysis with this dataset but they would need to at least cite it and if possible compare the slopes they used to this dataset (just an idea)

Similarly, the authors have not cited a (global) paper on shoreline retreat: Mentaschi, L., Vousdoukas, M. I., Pekel, J. F., Voukouvalas, E., & Feyen, L. (2018). Global long-term observations of coastal erosion and accretion. Scientific reports8(1), 12876. 

Following on the previous comments, I believe that the authors should extend the evaluation of their results with comparisons to these datasets (and possibly to local shoreline retreat numbers, if they can find any)

I find the numbers reported for shoreline retreat far too large. I know of very few beaches in e.g. the Eastern Mediterranean, that have a width of several tens of meters. I therefore find it strange that the authors report shoreline retreat of e.g. 100m, which in those environments would be possible only in soft cliffs (not beaches) or by submergence of low lying areas. Am i missing something? In this context, the authors could discuss the loss of features such as pocket beaches, which are important characteristics of Mediterranean coasts, or measures that would need to be carried out to prevent total loss of beaches.  

I am a little confused as to whether this approach over- or under- estimates erosion. In Line 535 it is claimed to underestimate but the opposite is written in line 562. Or am I missing something?  

Although the authors mention at several points that their results are important for decision making, they do not discuss how these could be used for decision making and at what level. I believe that the use of these results should be an important point in the discussion but seems to be missing at the moment.

Author Response

We thank the reviewers for their constructive comments and their very relevant suggestions. We acknowledge that while the framework for high-end sea-level scenarios was well developed in our manuscript (following Stammer et al., 2019 and other recent papers) the coastal erosion aspects were just at the state of the art (following Hinkel et al., 2014). Therefore, we updated the coastal erosion aspects using a recent database of beach slopes released by Athanasiou et al. (2019) in mid-October. These new aspects allow for more in depth analysis of the regional impacts of sea-level rise for shoreline changes in Europe, both for high-end scenarios and likely projections. We thank the reviewer for suggesting these changes, which we believe have strongly strengthened the paper.

In the following, our responses to the specific comments are preceded by bullet points. Line numbers in the revised version where changes have been made to respond to reviewer’s comments and suggestions are provided.

Reviewer #1

Comments and Suggestions for Authors

The manuscript discusses the development of regional sea-level projections for two high-end scenarios for Europe and, based on those scenarios, presents shoreline change projections for European sandy coasts. The manuscript is well written and the methods and results are nicely presented. Despite using simple methods, the authors have provided good justifications for their use and acknowledged and discussed the limitations. However I do have some concerns that would need to be addressed; and some suggestions which may help the authors improve the manuscript. Importantly, the authors have missed two very relevant references, need to extend the discussion on the implications of their results, and need to ensure access to the base data they are using. My points are listed below, in order of appearance (not importance) in the manuscript. I hope the authors find my comments and suggestions useful.

Although the development of the regional sea-level change projections occupies a large part of the manuscript and is one of the outputs of this paper, it is not mentioned in the abstract. I think that this point should also be included in the abstract.

As suggested by the reviewer, we now extended this point in the abstract (L17-21)

Line 52: why country scale in particular and not local or regional scale?  

In fact, our study focuses on regional scales (Europe to country level), although the results could be useful for local assessments too. Note, that there are already many papers that presented local results (e.g., work based on the PCR model – Ranasinghe et al., 2012). For sake of clarity, we removed “at country scale” (L 52).

Line 54: "do" instead of "does".

Done (L54).

Line 69: I am not sure what the authors mean by "...did not assess subsequent coastal impacts consistently...". Are they only referring to erosion?

Yes, in this case, we are referring to erosion. This has been clarified (L70).

Line 82: I think "only" should be removed 

Done (L83).

Line 121: "the" spatial coverage   "multi-model" not models 

Thank you: we added “the”, but did not understand where to add “multi” to “models” was necessary in our text. We did not see any ambiguity in this respect (L123).

Line 166: superimposing is probably not good enough as it would not account for non-linear effects. Maybe use a simpler term (e.g. considered)  

Done (L168)

I have not been able to access the Eurosion database. As the authors provide a link, this should provide direct access to the dbase. This is particularly important as there are issues with the database (which the authors do acknowledge) and therefore it should be clear where the input data come from. 

We agree with the reviewer. A link to access the database is now provided (L170-171).

Because of the those issues of accuracy and consistency with Eurosion, it might be useful to compare the dbase with other information to get an idea about its validity. To my knowledge, there are datasets available - e.g. DIVA, which the authors cite, local datasets (if available) or data from crowd sourcing from www.coastwards.org . I am not suggesting a full scale analysis here but just some comparisons to get an impression about the quality  

We agree with the reviewer that intercomparison of databases would be useful. However, we have three objections to performing this within the present study: first, we still compare our results with the results of DIVA. Second, while a comparison of databases discrepancies would be useful, this assessment goes well beyond the scope of this paper. In fact, the DIVA database is not publicly available (see DIVA webpage: https://globalclimateforum.org/portfolio-item/diva-model/#objectives). The framework for doing the assessment suggested by the reviewer could be the COASTMIP exercise (http://coastmip.org/). Finally, we thank the reviewer for suggesting the Coastward repository. However, this repository does not seem to be currently presented in the form of a GIS database, which will be necessary before being considered for intercomparison exercises such as COASTMIP.

Line 213: "global estimates are expected to be more accurate as they average across large areas". This is a statement which is repeated often by many and in very different contexts -  I find this rather misleading. Please explain or remove

We agree that the word “accurate” should be replaced by “precise” (L214-216): by averaging sea-level rise impacts over many shoreline positions, we obtain more precise results by assuming that errors due to lack of precise assessment of local sea-level changes or beach slopes are normally distributed around the “best” estimate. However, the results are not more accurate because we don’t know if the Bruun rule is accurate.

I find it a bit confusing that section 3 is separated from 2 as it seems to belong to methods, unless I have misunderstood something

We respectfully disagree with the reviewer here: section 2 describes the methods, while section 3 describes and justifies the framework; i.e. high-end scenarios. The methods provided in section 2 would remain relevant even under a different framework. In this regard, it appears more relevant to us to dissociate sections 2 and 3.

Very important: A paper providing coastal slope estimates globally has very recently been published: "Global distribution of nearshore slopes with implications for coastal retreat" Panagiotis Athanasiou et al. 2019 Earth System Science Data. I would assume that it would not be easy (or desirable) for the authors to repeat the analysis with this dataset but they would need to at least cite it and if possible compare the slopes they used to this dataset (just an idea)

We thank the reviewer for this indeed very important suggestion that led to a major reshaping of the study (see our introductory note). Including this new analysis using the nearshore slopes dataset in our study makes, we believe, our assessment sounder and more relevant.

Similarly, the authors have not cited a (global) paper on shoreline retreat: Mentaschi, L., Vousdoukas, M. I., Pekel, J. F., Voukouvalas, E., & Feyen, L. (2018). Global long-term observations of coastal erosion and accretion. Scientific reports8(1), 12876. 

We agree that this is a very relevant study. It is now included in the introduction (L79).  

Following on the previous comments, I believe that the authors should extend the evaluation of their results with comparisons to these datasets (and possibly to local shoreline retreat numbers, if they can find any)

In fact, the results from the recent observation-based studies (Luijendijk et al., 2018; Mentaschi et al., 2018) are not directly comparable to our results: these studies assess the current evolution of shorelines, which only include a small contribution from sea-level rise (as shown by Mentaschi et al 2018). Our results provide projections of impacts of sea-level rise for likely and high-end sea-level projections in Europe. A future study combining observed shoreline changes and future projections at global scale will be a worthwhile effort; it goes, however, well beyond the scope of the present study.

I find the numbers reported for shoreline retreat far too large. I know of very few beaches in e.g. the Eastern Mediterranean, that have a width of several tens of meters. I therefore find it strange that the authors report shoreline retreat of e.g. 100m, which in those environments would be possible only in soft cliffs (not beaches) or by submergence of low lying areas. Am i missing something? In this context, the authors could discuss the loss of features such as pocket beaches, which are important characteristics of Mediterranean coasts, or measures that would need to be carried out to prevent total loss of beaches.

As we now updated our analysis by including local nearshore slope characteristics, shoreline retreat rates have been considerably reduced in several regions and in Eastern Mediterranean in particular. Nonetheless, it is still likely that some number reported in some regions are too large because of the initial assumptions –e.g. infinite erosion potential, no adaptation measure, … - we made to conduct this study (but which are necessary as our assessment is done at a very large scale). This constitutes obvious limitations that we now emphasize more clearly in the discussion section (L588-617).

I am a little confused as to whether this approach over- or under- estimates erosion. In Line 535 it is claimed to underestimate but the opposite is written in line 562. Or am I missing something?  

We agree that this may be confusing, although these statements are actually not inconsistent as they reflect different scale analysis and refer to different effects. Now (L582 in the revised manuscript), we clarified that both pan-European assessments (ours and DIVA), when using a 1% nearshore slope, likely underestimate the land loss area potential. This is because in our case, we do not include tidal basins and in the DIVA case, the global length of sandy shoreline is likely underestimated. This conclusion arises from the comparison of both assessments and their modeling framework (L565-582). In the revised version of our assessment, the use of observation-based nearshore slopes further supports the fact that 1% uniform nearshore slopes-based assessments likely underestimate land loss area potential at European scale in general. This is also now clarified. Regarding the second paragraph (L601-617), we refer to other limitations of our modeling approach (see also previous comment) and in particular, to the fact that we assume that beaches can erode infinitely. For countries where the backshore erodibility is weak – such as Greece or United Kingdom – it is obvious that the infinite beach assumption is not very relevant and in this case, our land loss projections are likely overestimated (but this does not hold for all European countries). This has been clarified (L586-591). In any case, any uncertainty of our shoreline retreat modeling framework can potentially alter our projections in either direction (under – or – over estimation), but a in-depth and quantitative evaluation of each contribution to land loss uncertainty is far beyond the scope of our study.

Although the authors mention at several points that their results are important for decision making, they do not discuss how these could be used for decision making and at what level. I believe that the use of these results should be an important point in the discussion but seems to be missing at the moment.

In our study, we provide an assessment which can be useful to decision making – but examining all implications for decision making (incl. for decision makers such as European investment funds supporting adaptation) is another study. We just provide elements that may support such study. The statement that high-ends are suited for robust decision making and users with low tolerance of uncertainties is not part of our work but results from the studies Hinkel et al (2019) and Stammer et al (2019). We acknowledge this was not always clear in the previous version of the manuscript and now include more explicit references to Hinkel et al. (2019) and Stammer et al. (2019) in the discussion part (L533, L619-620). In the discussion, note also that potential implications of our findings on adaptation strategy are mentioned at several places.

Reviewer 2 Report

The paper describes 1) high-end sea level rise scenarios and 2) the potential impacts of the sea level rise on sandy coastlines.

The article is well written and exposes very valuable results on regional sea level scenarios. The part of this paper discussing impacts needs however some corrections detailed below.

My suggestion is therefore to recommend this paper for publication after corrections can be made.

Detailled comments :

-The Bruun rule can also be viewed as a simple trigonometric formula. Shoreline recession is a simple geometric result of sea level rise, for any sloping coastline. This does not imply erosion, or any sediment movements. Sediment movements can increase (erosion) or mitigate (accretion) the shoreline recession caused by sea level rise. The authors appear to equate shoreline recession with erosion at several instances in the text. My recommendation is to only mention shoreline recession, given the scope of the paper.

-The authors appropriately mention that the Bruun rule applicability has been severely questioned. I agree that in their approach, the authors make the case for using it anyway.

The authors chose to use a constant beach slope. My recommendation here is to discuss the sensitivity of their results to their beach slope choice, in term of shoreline recession order of magnitude.

Among other parameters, the beach slope is dependent on wave characteristics (see for example Sunamura 1984, or other articles linking wave characteristics to equilibrium beach slope). A discussion of the potential aggravating or mitigating effects of wave climate change (using Bricheno et al. 2018 for example) on their results would also be useful.

Sunamura, T. Quantitative predictions of Beach face slopes. Geol.Soc.Am.Bull.95,242–245 (1984).

Bricheno, L. M., & Wolf, J. ( 2018). Future wave conditions of Europe, in response to high‐end climate change scenarios. Journal of Geophysical Research: Oceans, 123, 8762– 8791. https://doi.org/10.1029/2018JC013866

Author Response

We thank the reviewers for their constructive comments and their very relevant suggestions. We acknowledge that while the framework for high-end sea-level scenarios was well developed in our manuscript (following Stammer et al., 2019 and other recent papers) the coastal erosion aspects were just at the state of the art (following Hinkel et al., 2014). Therefore, we updated the coastal erosion aspects using a recent database of beach slopes released by Athanasiou et al. (2019) in mid-October. These new aspects allow for more in depth analysis of the regional impacts of sea-level rise for shoreline changes in Europe, both for high-end scenarios and likely projections. We thank the reviewer for suggesting these changes, which we believe have strongly strengthened the paper.

In the following, our responses to the specific comments are preceded by bullet points. Line numbers in the revised version where changes have been made to respond to reviewer’s comments and suggestions are provided.

Reviewer #2

Comments and Suggestions for Authors 

The paper describes 1) high-end sea level rise scenarios and 2) the potential impacts of the sea level rise on sandy coastlines.

The article is well written and exposes very valuable results on regional sea level scenarios. The part of this paper discussing impacts needs however some corrections detailed below.

My suggestion is therefore to recommend this paper for publication after corrections can be made.

 Detailled comments :

-The Bruun rule can also be viewed as a simple trigonometric formula. Shoreline recession is a simple geometric result of sea level rise, for any sloping coastline. This does not imply erosion, or any sediment movements. Sediment movements can increase (erosion) or mitigate (accretion) the shoreline recession caused by sea level rise. The authors appear to equate shoreline recession with erosion at several instances in the text. My recommendation is to only mention shoreline recession, given the scope of the paper.

We agree with this comment. We made the necessary changes in the text wherever it was relevant, including the title. Nonetheless, we note that shoreline recession further includes potential permanent inundation, which we did not account for. This has been clarified in the introduction (L88-90) and reminded in the discussion (L617-621) to remove any possible ambiguity.   

-The authors appropriately mention that the Bruun rule applicability has been severely questioned. I agree that in their approach, the authors make the case for using it anyway.

We agree with the reviewer. Nevertheless, we also acknowledge that (1) the Bruun rule cannot be ruled out yet, (2) even if the physics of the Bruun rule is wrong, there are other conceptual models that came up with the same formula (e.g., https://www.jcronline.org/doi/full/10.2112/03-0051.1), (3) the Bruun rule appears as a high end estimate of shoreline change projections based on the alternative PCR model (Ranasinghe et al., 2012). For these reasons, we argue that using the Bruun rule as part of a high-end estimate of sea level rise impacts on shoreline changes is a consistent approach. In the revised version of the manuscript, we extended the discussion on the Bruun rule limitation in the last section.

The authors chose to use a constant beach slope. My recommendation here is to discuss the sensitivity of their results to their beach slope choice, in term of shoreline recession order of magnitude.

We thank the reviewer for this comment. Based on this comment and on the other reviewer’s suggestions, we now compare results using a nearshore slope dataset (L219-232) or a 1% uniform slope approach (as in the previous version of the manuscript) and discuss the differences and the consequences on shoreline changes projection (section 4 and 5).

Among other parameters, the beach slope is dependent on wave characteristics (see for example Sunamura 1984, or other articles linking wave characteristics to equilibrium beach slope). A discussion of the potential aggravating or mitigating effects of wave climate change (using Bricheno et al. 2018 for example) on their results would also be useful.

We agree with reviewer. Note that our results show that considering the structural uncertainties pertaining the Bruun rule and future sea-level projections provides already a very wide range of SLR-induced shoreline change projections. Hence, we mentioned the potential additional source of uncertainty due to wave-induced beach slope changes and cite the relevant references in the discussion (L603), but we remained brief.

Sunamura, T. Quantitative predictions of Beach face slopes. Geol.Soc.Am.Bull.95,242–245 (1984).

Bricheno, L. M., & Wolf, J. ( 2018). Future wave conditions of Europe, in response to high‐end climate change scenarios. Journal of Geophysical Research: Oceans, 123, 8762– 8791. https://doi.org/10.1029/2018JC013866

Round 2

Reviewer 1 Report

I have now read the revised manuscript and the authors' response. I feel that the authors have addressed all comments in a detailed manner and that the manuscript has improved. I find it commendable that the authors actually used the new dataset in their analysis.

I believe that the manuscript should be published after one minor change: the link for the EUROSION dataset still does not work! The authors should definitely fix this before the manuscript is published.

Two more quick points (I leave it to the discretion of the authors to address those if they want):

to clarify my previous comment, now line 133, I believe it should be "multi-model statistics" instead of "multi-modelS statistics" Regarding my suggestions for doing a quick qualitative comparison of the EUROSION dbase with other sources, there is a version of DIVA for the Mediterranean that includes a detailed coastal morphology (Wolff et al., 2018, Scientific Data) and the data from Coastwards can be downloaded with their X,Y co-ordinates (and can therefore be easily imported in GIS).

Once again, only the first point (link to EUROSION) needs to be addressed, the rest is up to the authors.
I hope the authors have found my comments somehow useful for improving the manuscript.

Reviewer 2 Report

By answering a simple criticism, the authors provided an detailled analysis of the effect of beach slope on their results. It greatly improves the mansucript and provides a nice balance between the 2 main parts of the paper .

All other concerns were answered. 

I recommend to accept this paper.

Author Response

We thank the reviewer for her/his very useful suggestions throughout the review process that contributed to improve greatly our manuscript.